# Lymphopenia in hospitalized patients and its relationship with severity of illness and mortality

Juan Carlos Andreu-Ballester[1,2,3,4,5]*, Aurelio Pons-Castillo[2], Antonio González-Sánchez[3], Antonio Llombart-Cussac[4], María José Cano[5], Carmen Cuéllar[6]

1 Research Department, Arnau de Vilanova-Lliria University Hospital, Valencia, Spain, 2 Clinical Analysis Department, Arnau de Vilanova-Lliria University Hospital. Valencia, Spain, 3 Clinical Documentation Department, Arnau de Vilanova-Lliria University Hospital. Valencia, Spain, 4 Medical Oncology Department, Arnau de Vilanova-Lliria University Hospital, Valencia, Spain, 5 Emergency Department, Arnau de Vilanova-Lliria University Hospital, Valencia, Spain, 6 Department of Microbiology and Parasitology. Complutense University, Madrid, Spain

* jcandreuballester@outlook.com

## Abstract

### Background

Lymphopenia is associated with various pathologies such as sepsis, burns, trauma, general anesthesia and major surgeries. All these pathologies are clinically expressed by the so-called Systemic Inflammatory Response Syndrome which does not include lymphopenia into defining criteria. The main objective of this work was to analyze the diagnosis of patients admitted to a hospital related to lymphopenia during hospital stay. In addition, we investigated the relationship of lymphopenia with the four levels of the Severity of Illness (SOI) and the Risk of Mortality (ROM).

### Method and findings

Lymphopenia was defined as Absolute Lymphocyte Count (ALC) <1.0 x$10^9$/L. ALC were analyzed every day since admission. The four levels (minor, moderate, major and extreme risk) of both SOI and ROM were assessed. A total of 58,260 hospital admissions were analyzed. More than 41% of the patients had lymphopenia during hospital stay. The mean time to death was shorter among patients with lymphopenia on admission 65.6 days (CI95%, 57.3–73.8) vs 89.9 (CI95%, 82.4–97.4), P<0.001. Also, patients with lymphopenia during hospital stay had a shorter time to the mortality, 67.5 (CI95%, 61.1–73.9) vs 96.9 (CI95%, 92.6–101.2), P<0.001.

### Conclusions

Lymphopenia had a high prevalence in hospitalized patients with greater relevance in infectious pathologies. Lymphopenia was related and clearly predicts SOI and ROM at the time of admission, and should be considered as clinical diagnostic criteria to define SIRS.

**Data Availability Statement:** All relevant data are within the manuscript and its Supporting Information files.

**Funding:** The author(s) received no specific funding for this work.

**Competing interests:** The authors have declared that no competing interests exist.

## Introduction

Lymphopenia is associated with various pathologies such as sepsis, burns, trauma, general anesthesia and mayor surgeries. This leads to a life-threatening period of immunosuppression [1–5]. This situation represents a serious public health problem, since both incidence and mortality from these processes are increasing [6–8]. All these pathologies have in common the clinical expression of the so-called Systemic Inflammatory Response Syndrome (SIRS), which includes a number of clinical and laboratory criteria such as leukocytosis and leukopenia, but does not include lymphopenia [9].

There is a need for a consensual definition of SIRS and sepsis, since there is sometimes difficulty, with the current criteria, to define pathological cases with SIRS caused by infection, since it is not easy to identify the infectious focus in some situations. Therefore, those criteria should be better adjusted to avoid erroneous diagnoses. Persistent lymphopenia is an independent predictor of increased mortality in critically ill emergency general surgical patients [10], septic patients [11], and outpatients [12, 13]. However, to our knowledge, lymphopenia has not been studied in all the pathologies that make up the repertoire of diagnoses of patients admitted to hospital.

The All Patients Refined Diagnosis Related Groups (APR-DRGs) are a patient classification scheme by means of several attributes which include severity of illness, risk of dying, prognosis, treatment difficulty, need for intervention, and resource intensity. Severity of Illness (SOI) refers to the extent of physiologic decompensation or organ system and Risk of Mortality (ROM) refers to the likelihood of dying. There are four SOI and ROM subclasses which are numbered sequentially from 1 to 4 indicating respectively, minor (1), moderate (2), major (3), or extreme (4) severity of illness or risk of mortality [14–16].

The main objective of this work was to analyze the diagnosis of patients admitted to hospital related to lymphopenia on admission and during hospital stay. In addition, we investigated the relationship of lymphopenia with the four levels of SOI and ROM according to DRGs.

## Methods

### Study design and obtaining information

This retrospective cohort study was conducted at the Arnau de Vilanova–Llíria Health Department in Valencia, Spain, including two hospitals: Arnau de Vilanova's Hospital and Llíria's Hospital, with a total of 450 beds, which cover an area of 300,000 inhabitants. Patients older than 14 years were admitted from January 1, 2016, to December 31, 2019, from Emergency Department (80%) and Scheduled Admission (20%). A file was generated with all analytical parameters of patients seen in the emergency room and hospitalization in the four years of the study. A cross was made with hospital episodes and emergency data. The resulting file was crossed, in turn, with the Database of DRGs from those same years. We obtained a Database of 72,984 records that combines data related to the hospitalization episode (date of admission, date of discharge, reason for admission, diagnoses, procedures); data related to the patient (age, sex); data related to DRGs (DRG, CMD, Severity, Risk of Mortality). Finally, patients who had undergone in-hospital laboratory tests were selected, ordered by evolution of the laboratory results in the admission episode, obtaining a sample of 58,860 patients. Clinical diagnoses were coded using International Classification of Diseases (ICD-10, 2016 version). The study protocol was approved by the Ethics and Investigation Committee of the Arnau de Vilanova-Llíria Hospital (approval number 2019–12), Valencia City, Spain.

### Study variables

**Demographic.** Age and gender. Hospital stay, up to 15 diagnoses associated with each patient and at each admission, all blood counts performed in each episode, and in-hospital mortality.

Lymphopenia was defined as Absolute Lymphocyte Count (ALC) <1.0 x10$^9$/L. ALC were analyzed every day from admission. The mean of the analyses performed each day of hospitalization was calculated. We assessed the Severity of Illness (SOI) and the Risk of Mortality (ROM), with its four levels, minor (1), moderate (2), major (3), and extreme risk (4).

### Statistical analysis

When normality was assumed (Kolmogorov-Smirnov test), Student *t* test was used to compare the means between two different groups of patients. When the hypothesis of normality was not accepted, the Mann-Whitney U test was used. Two-tailed Fischer's exact test was used to compare lymphopenia with diagnosis (Odds Ratio). Logistic regression (Exp B) and Cox regression (Hazard Ratios) were used to compare the two groups (with lymphopenia and without lymphopenia) taken into account analytical variables. Repeated Measures ANOVA were used to assess the differences of lymphocytes frequency between dead and alive patients over time. The Kaplan-Meier survival method was used to estimate the cumulative probability of mortality according to lymphopenia on admission and lymphopenia during hospital stay. Differences between curves were tested using the Log-Rank test (Statistical program R, version 3.3). A Cox proportional hazards regression model was used to assess whether lymphopenia variables were independently associated with the risk of mortality. P value <0.05 was considered statistically significant.

## Results

### Characteristics of admitted patients and diagnosis

A total of 58,260 hospital admissions were analyzed. The mean age was 67.8 ± 18.4 years (66.4 ± 17.7 in males and 69.5 ± 19.0 in females, P<0.001). The mean age of patients with lymphopenia was 73.5 ± 15.6 years *vs* 63.9 ± 19.2 in patients without lymphopenia, p<0.001.

The mean of hospital stay was 6.5 ± 6.6 days (6.8 ± 6.9 in males and 6.1 ± 6.1 in females, p <0.001). The mean of hospital stay in lymphopenic patients was 7.7 ± 7.6 days *vs* 5.6 ± 5.7 days in no lymphopenic patients (*p*<0.001). A total of 23,892 (41.0%) patients presented lymphopenia during hospital stay. Eighteen thousand three patients (30.9%) presented lymphopenia on admission. The characteristics of the admitted patients and the relationship of the variables studied according to lymphopenia are shown in Table 1.

### Relationship of lymphopenia with diagnosis

Table 2 shows the relationship of lymphopenia according to Major Diagnostic Categories (MDC).

The relationships between lymphopenia and diagnostic categories were analyzed using univariate analysis. International Statistical Classification of Diseases and Related Health Problems (ICD-10) was used. The results are shown in S1 Table.

Infectious processes were the most frequently coded (21,093, 36.2%). "In patients without infectious pathology, the incidence of lymphopenia was 34.6% (*n* = 12,876/37,167) vs 52.2% in patients with infectious pathology (*n* = 11,016/21,093); OR$_{lymphopenia\ without\ infectious\ pathology}$ = 0.66 (0.63–0.65), p<0.001 " OR$_{lymphopenia\ with\ infectious\ pathology}$ = 2.06 (1.99–2.14), p<0.001".

There were 6,208/58,260 (10.7%) patients with sepsis, 3,528 (56.8%) of which presented lymphopenia vs 2,680 (43.2%) who had lymphopenia in the group without sepsis, OR = 2.1 (1.9–2.2), p<0.001.

**Table 1. Characteristics of admitted patients and their relationship with lymphopenia (univariant analysis).**

| | Cases | Lymphopenia | OR (CI 95%) | Sig. p |
|---|---|---|---|---|
| | $n$ = 58,260 | $n$ = 23,892 (41.0%) | | |
| | $n$ (%) | $n$ (%) | | |
| **Gender** | | | | |
| Male | 30,597 (52.5) | 12,987 (42.4) | 1.13 (1.09–1.17) | < 0.001 |
| Female | 27,663 (47.5) | 10,905 (39.4) | 0.88 (0.85–0.91) | < 0.001 |
| **Origin of Hospital Admission** | | | | |
| Emergency Admission | 46,598 (80.0) | 19,891 (42.7) | 1.43 (1.37–1.49) | < 0.001 |
| Scheduled Admission | 11,662 (20.0) | 4,001 (34.3) | 0.70 (0.67–0.73) | < 0.001 |
| **Diagnosis Related Groups** | | | | |
| Surgical APR-DRG | 11,820 (20.3) | 5,094 (43.1) | 1.11 (1.07–1.16) | < 0.001 |
| Medical APR-DRG | 46,440 (79.7) | 18,798 (40.5) | 0.90 (0.86–0.94) | < 0.001 |
| **Severity of Illness** | | | | |
| SOI-1 | 22,329 (38.3) | 6,613 (29.6) | 0.45 (0.44–0.47) | < 0.001 |
| SOI-2 | 26,583 (45.6) | 11,489 (43.2) | 1.18 (1.14–1.22) | < 0.001 |
| SOI-3 | 8,788 (15.1) | 5,348 (60.9) | 2.59 (2.48–2.72) | < 0.001 |
| SOI-4 | 560 (1.0) | 442 (78.9) | 5.47 (4.46–6.71) | < 0.001 |
| **Risk of Mortality** | | | | |
| ROM-1 | 30,052 (51.6) | 8,553 (28.5) | 0.33 (0.32–0.35) | < 0.001 |
| ROM-2 | 19,449 (33.4) | 9,821 (50.5) | 1.79 (1.73–1.86) | < 0.001 |
| ROM-3 | 7,674 (13.2) | 4,705 (61.3) | 2.59 (2.47–2.72) | < 0.001 |
| ROM-4 | 1,085 (1.9) | 813 (74.9) | 4.42 (3.85–5.07) | < 0.001 |
| **Mortality** | 3,213 (5.5) | 2,345 (73.0) | 4.20 (3.88–4.55) | < 0.001 |

APR-DRG: Refined Diagnosis Related Group. SOI: Severity of Illness. ROM: Risk of Mortality. Significance: Two-tailed Fischer's exact test was used.

The patients with Septic Shock were 515/58.260 (0.9%), 446 (86.6%) of which had lymphopenia. Only 69 (13.4%) patients with septic shock did not have lymphopenia, OR = 9.5 (7.3–12.2), p<0.001.

S1 Appendix shows the number of diagnoses in each patient according to disease groups.

Each patient can have multiple diagnoses, so there are more diagnoses than patients. S1 Appendix shows the number of patients who have one or more diagnoses at the same time according to disease groups. Finally, it can be seen that the sum of the totals is equal to the sum of the diagnoses.

## Mortality and lymphopenia

The number of deaths was 3,213 (5.5%). Of these patients, 2,345 (73.0%) had lymphopenia during hospital stay and 1,743 (54.2%) had lymphopenia on admission, OR = 4.2 (CI95%, 3.9–4.6) and OR = 2.8 (CI95%, 2.5–3.1) respectively, P<0.001. Fig 1 shows the evolution of absolute lymphocytes counts (mean) according to alive and dead patients in the consecutive days analyzes performed (1 to 20 days) during the hospital stay, in four situations: all cases, infectious pathology, sepsis and septic shock. OA: On Admission. Lymphopenia—<1.0 x$10^9$/L. Repeated Measures ANOVA (p<0.05). T-bars show typical error. Significative differences (Mann-Whitney U test) between alive and dead: Total patients; Infectious Diseases and Sepsis ($p$<0.001); Septic Shock: days 1 to 10 ($p$<0.001), days 11 to 15 ($p$<0.05); days 16 to 20 no significant.

The Kaplan-Meier curve for the risk of mortality is shown in Fig 2. Kaplan-Meier plots illustrating Survival probability following lymphopenia on admission **(A)** and lymphopenia

**Table 2. Relationship of lymphopenia according to Major Diagnostic Categories (MDC) (*n* = 58,260).**

| | Cases | Lymphopenia *n* = 23,892 (41.0%) | OR (CI 95%) | Sig. p |
|---|---|---|---|---|
| | *n* | *n* (%) | | |
| 01 - Nervous System | 3,823 | 954 (25.0) | 0.46 (0.42–0.49) | < 0.001 |
| 02 - Eye | 216 | 30 (13.9) | 0.23 (0.16–0.34) | < 0.001 |
| 03 - Ear, Nose, Mouth and Throat | 965 | 411 (42.6) | 1.07 (0.94–1.22) | NS |
| **04 - Respiratory System** | **10,530** | **5,658 (53.7)** | **1.88 (1.80–1.96)** | < 0.001 |
| 05 - Circulatory System | 8,868 | 3,003 (33.9) | 0.70 (0.67–0.73) | < 0.001 |
| 06 - Digestive System | 7,461 | 2,952 (39.6) | 0.93 (0.89–0.98) | 0.007 |
| **07 - Hepatobiliary System and Pancreas** | **3,311** | **1,685 (50.9)** | **1.53 (1.42–1.64)** | < 0.001 |
| **08 - Musculoskeletal System and Connective Tissue** | **5,950** | **2,725 (45.8)** | **1.24 (1.18–1.31)** | < 0.001 |
| 09 - Skin, Subcutaneous Tissue and Breast | 1,006 | 320 (31.8) | 0.67 (0.58–0.76) | < 0.001 |
| 10 - Endocrine, Nutritional and Metabolic System | 1,312 | 463 (35.3) | 0.78 (0.70–0.86) | < 0.001 |
| **11 - Kidney and Urinary Tract** | **5,293** | **2,285 (43.2)** | **1.10 (1.04–1.17)** | 0.001 |
| 12 - Male Reproductive System | 749 | 262 (35.0) | 0.77 (0.66–0.90) | 0.001 |
| 13 - Female Reproductive System | 1,341 | 246 (18.3) | 0.32 (0.28–0.36) | < 0.001 |
| 14 - Pregnancy, Childbirth and Puerperium | 9 | 2 (22.2) | 0.41 (0.85–1.98) | NS |
| **16 - Blood and Blood Forming Organs and Immunological Disorders** | **946** | **529 (55.9)** | **1.84 (1.62–2.10)** | < 0.001 |
| **17 - Myeloproliferative Diseases (Poorly Differentiated Neoplasms)** | **863** | **540 (62.6)** | **2.44 (2.10–2.80)** | < 0.001 |
| **18 - Infectious and Parasitic Diseases** | **1,575** | **1,031 (65.5)** | **2.80 (2.52–3.12)** | < 0.001 |
| 19 - Mental | 1,652 | 162 (9.8) | 0.15 (0.13–0.18) | < 0.001 |
| 20 - Alcohol/Drug Use or Induced Mental Disorders | 927 | 75 (8.1) | 0.12 (0.10–0.16) | < 0.001 |
| 21 - Injuries, Poison and Toxic Effect of Drugs | 439 | 151 (34.4) | 0.75 (0.62–0.98) | 0.005 |
| 23 - Factors Influencing Health Status and Other Contacts Health Services | 803 | 282 (35.1) | 0.78 (0.67–0.90) | 0.004 |
| **24 –Human Immunodeficiency Virus Infection** | **197** | **112 (56.9)** | **1.90 (1.43–2.52)** | < 0.001 |
| **25 - Multiple Significant Trauma** | **22** | **14 (63.6)** | **2.52 (1.06–6.00)** | 0.048 |

The diagnoses in each MDC correspond to a single organ system or cause, fundamentally based on the main diagnosis on admission. Therefore, there is only one MDC for each hospitalization episode.

during hospital stay **(B)** in peripheral blood of admitted patients. Time is expressed in days (mean). Log-Rank test (p-value). Percentage of dead patients according to the number of analyzes performed with lymphopenia on admission or during admission **(C)**. Percentage of patients with lymphophenia on admission and during hospital stay according to four subclases of Severity of illness (SOI) **(D)** and Risk of Mortality (ROM) **(E)**. Chi square Pearson test, p<0.001. Pearson's R: Interval-by-interval, p<0.001. Spearman correlation: Ordinal-by-ordinal, p<0.001 (D and E panels). The mean time to death was shorter among lymphopenia on admission (A) 65.6 days (CI95%, 57.3–73.8) *vs* 89.9 (CI95%, 82.4–97.4), P< 0.001. In addition, lymphopenia during hospital stay in hospital admission patients (B) had a shorter time to the mortality, 67.5 (CI95%, 61.1–73.9) *vs* 96.9 (CI95%, 92.6–101.2), p< 0.001.

Furthermore, patients with lymphopenia on admission had a higher risk of mortality on a multivariate analysis, Hazard Ratio (HR) adjusted: 2.4 (CI95%, 2.2–2.5), P<0.001 and lymphopenia during hospital stay, HR adjusted: 2.8 (CI95%, 2.6–3.0) P<0.001. The percentage of patients who die increases progressively according to the number of analyzes performed with lymphopenia during their hospital admission (Fig 2C). Relationship of lymphopenia with severity and mortality risk has been described in Fig 2, panel D and E, respectively.

Table 3 shows relationship between leukocyte and platelet counts with mortality rates.

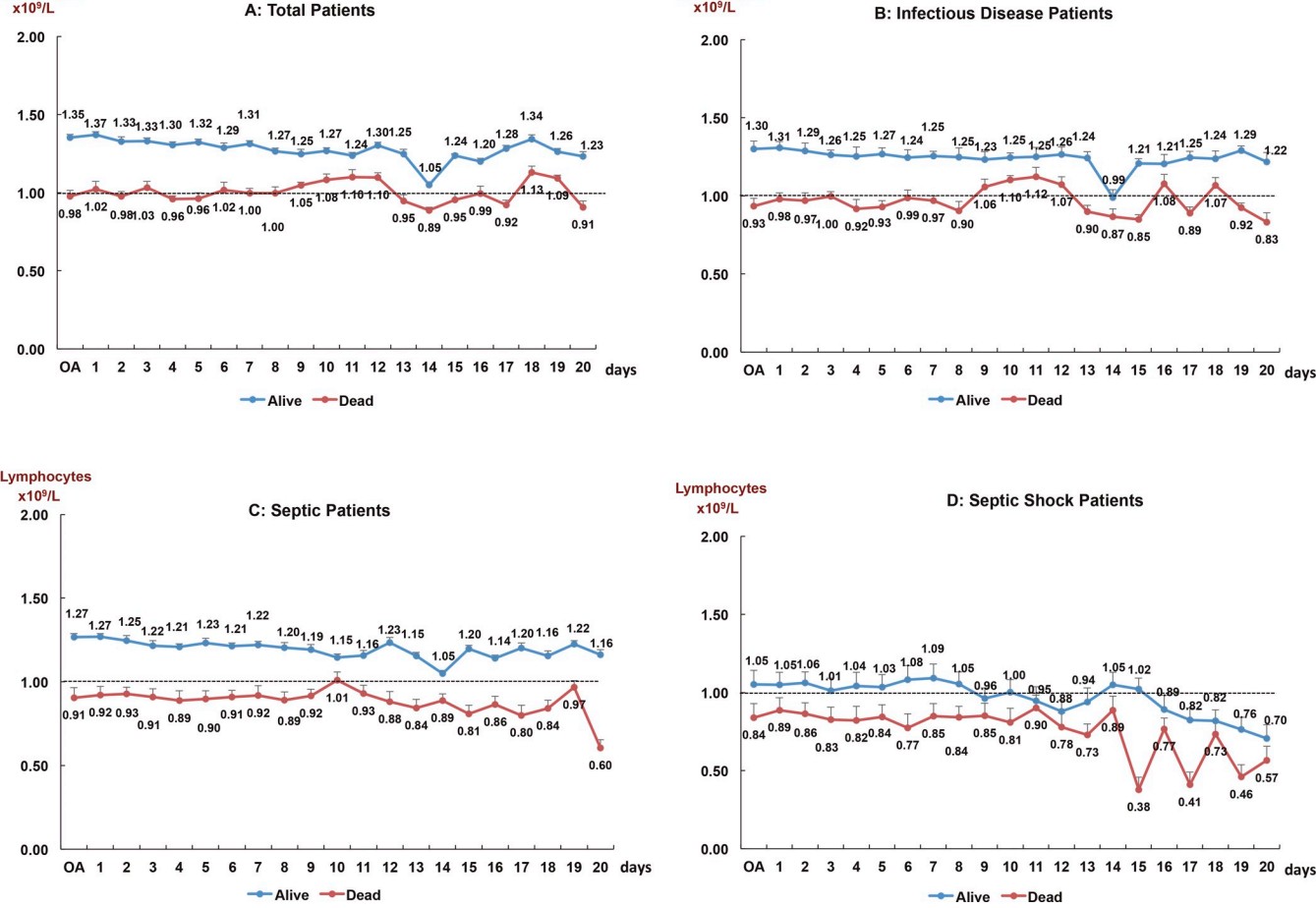

**Fig 1. Evolution of absolute lymphocytes counts (mean) according to alive and dead patients in the consecutive days analyzes performed (1 to 20 days) during the hospital stay, in four situations: All cases, infectious pathology, sepsis and septic shock.** OA: On Admission. Lymphopenia—<1.0 x10⁹/L. Repeated Measures ANOVA (p<0.05). T-bars show typical error. Significative differences (Mann-Whitney U test) between alive and dead: Total patients; Infectious Diseases and Sepsis ($p<0.001$); Septic Shock: days 1 to 10 ($p<0.001$), days 11 to 15 ($p<0.05$); days 16 to 20 no significant.

## Discussion

This is the first study to evaluate the association of lymphopenia with all hospital diagnoses in such a large number of patients. Highlights the high prevalence of lymphopenia in our study, since more than 30% of the patients admitted to the hospital had lymphopenia on admission. In addition, the incidence increased to 41% when including lymphopenia during hospital stay.

Lymphopenia was more common in men than in women. On the other hand, lymphopenia was higher among patients admitted for surgical pathologies *vs* medical processes and it was also more frequent in Emergency compared to Scheduled Admission.

The greatest decrease in lymphocytes was observed in pathological processes associated with infection. The lowest values were detected in sepsis and septic shock [17, 18]. The most probable cause of this transient phenomenon of immunoparalysis with decreased of lymphocytes has been attributed to increased apoptosis of this cells [19, 20]. Considering the importance of lymphocytes in the host's immune defense, it is not surprising the relationship of lymphopenia with severity and mortality. This fact was previously observed in other studies, although in a smaller number of patients. A prospective Danish population-based study, in subjects without acute pathology and underwent voluntary analysis, demonstrated an

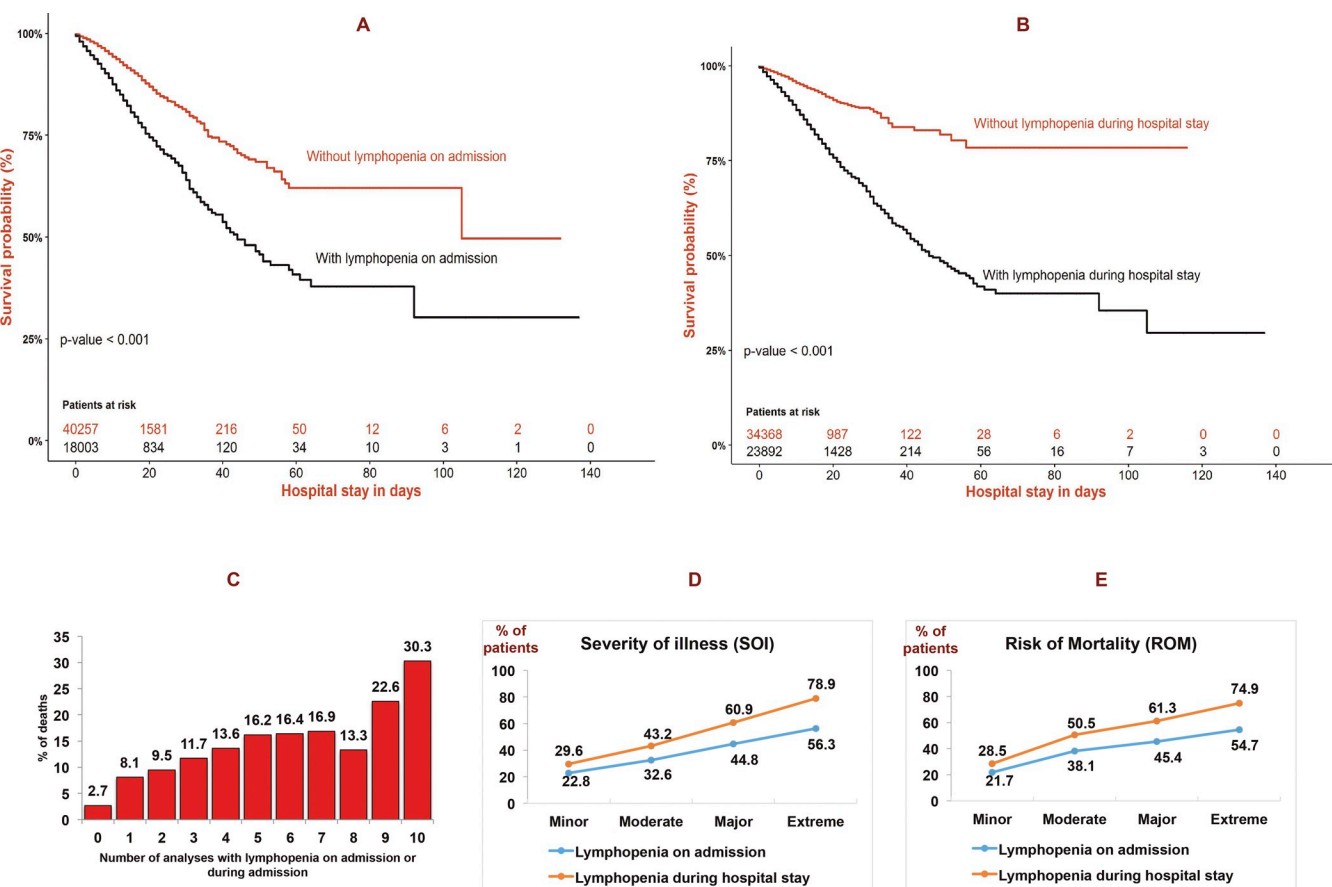

**Fig 2.** Kaplan-Meier plots illustrating Survival probability following lymphopenia on admission **(A)** and lymphopenia during hospital stay **(B)** in peripheral blood of admitted patients. Time is expressed in days (mean). Log-Rank test (*p*-value). Percentage of dead patients according to the number of analyzes performed with lymphopenia on admission or during admission **(C)**. Percentage of patients with lymphopenia on admission and during hospital stay according to four subclases of Severity of illness (SOI) **(D)** and Risk of Mortality (ROM) **(E)**. Chi square Pearson test, p<0.001. Pearson's R: Interval-by-interval, p<0.001. Spearman correlation: Ordinal-by-ordinal, p<0.001 (D and E panels).

association between lymphopenia and increased risk of hospitalization and infection-related death [13]. We previously observed that lymphopenia was present in 75% of patients admitted to the hospital with sepsis. An inverse relationship of γδ T cells with disease severity and mortality was observed in septic patients [21].

In this work, a significant direct relationship between SOI and ROM with lymphopenia has been demonstrated. SOI and ROM variables are not available for the clinician at the time of admission. However, lymphocyte count and other objective parameters are available at the time of admission. The definition of SIRS is based on clinical and analytical criteria easily obtained and quickly accessible to clinicians. Leukocytes and especially total neutrophils and "band" neutrophils are routinely performed in the emergency laboratory. The interpretation of leukocytosis and leukopenia is used as criteria for SIRS and infection. In this work we showed that lymphopenia has a higher predictive value for mortality than the measurement of leukocytes and total neutrophils.

The inclusion criteria for SIRS did not allow distinguishing uninfected patients from septic patients. However, as we demonstrated in our study, lymphopenia is clearly related to infectious pathology, being much more marked in sepsis and in septic shock [9].

**Table 3. Relationship between leukocyte and platelet counts with mortality rates.**

| | Cases | Deaths | | | | | | | | |
|---|---|---|---|---|---|---|---|---|---|---|
| | n (%) | n | Fatality rate | % in Total Deaths | Two-tailed Fischer's exact test | Sig. | Logistic Regression | Sig. | Cox Regression | Sig. p |
| | 58,260 (100.0) | 3,213 | (%) (5.5%) | 100 | | | Exp (B) (CI95%) | | HR Adjusted | |
| Lymphopenia (< 1.0 x 10⁹/L) | 23,892 (41.0) | 2,345 | (9.8) | (73.0) | 4.2 (3.9–4.5) | <0.001 | 2.84 (2.63–3.07) | <0.001 | 2.30 (2.12–2.50) | <0.001 |
| Neutrophylia (> 7.0 x 10⁹/L) | 33,570 (57.6) | 2,631 | (7.8) | (81.9) | 3.5 (3.2–3.9) | <0.001 | 2.74 (2.50–3.00) | <0.001 | 1.75 (1.56–1.96) | <0.001 |
| Leukocitosis (>12.1 x 10⁹/L) | 21,433 (36.8) | 1,973 | (9.2) | (61.4) | 2.9 (2.7–3.1) | <0.001 | 2.19 (2.03–2.35) | <0.001 | 1.49 (1.33–1.58) | <0.001 |
| Thrombocytopenia (< 125 x 10⁹/L) | 7,470 (12.8) | 811 | (10.9) | (25.2) | 2.5 (2.2–2.7) | <0.001 | 1.66 (1.53–1.58) | <0.001 | 1.49 (1.37–1.63) | <0.001 |
| Leukopenia (< 4.0 x 10⁹/L) | 3,818 (6.6) | 292 | (7.6) | (9.1) | 1.5 (1.3–1.7) | <0.001 | 0.98 (0.86–1.10) | 0.977 | 0.79 (0.67–0.94) | 0.006 |
| Neutropenia (<1.4 x 10⁹/L) | 1,590 (2.7) | 143 | (9.0) | (4.5) | 1.7 (1.5–2.1) | <0.001 | 1.05 (0.88–1.24) | 0.152 | 1.07 (0.86–1.34) | 0.536 |

All analytical variables were included in the analysis using logistic regression and Cox regression.

A greater association was observed between mortality and lymphopenia during hospitalization than with lymphopenia on admission. This finding could be due to the fact that the percentage of patients who die increased progressively according to the number of analyzes performed during their hospitalization.

In summary, the decrease in lymphocytes should be valued by physicians as a factor to consider in the prognosis of their patients. In conclusion, lymphopenia should be included among the criteria that evaluate the SOI and ROM at the time of admission. Furthermore, lymphopenia should be a criterion for evaluating SIRS patients with infectious and non-infectious causes at the time of admission.

## Limitations

Our study has the limitations of retrospective studies. In addition, it lacks pediatric patients and patients with large burns since these pathologies are centralized in another hospital center whose database we have not been able to access. In any case, we think that the number of patients evaluated in the present study is large enough to be able to draw solid conclusions.

## Conclusion

Lymphopenia has a high prevalence in patients admitted to a general hospital, with greater relevance in subjects with infectious pathology. Furthermore, lymphopenia is related and clearly predicts the Severity of Illness and the Risk of Mortality at the time of admission, and should be considered as a clinical diagnostic criteria to define SIRS patients with infectious and non-infectious causes at the time of admission.

## Supporting information

**S1 Table. Significant association lymphopenia according to diagnostic (CIE-10)**
**(n = 58,260).**
(DOCX)

**S2 Table. Significant association lymphopenia according to infectious diseases (CIE-10).** (DOCX)

**S1 Appendix. Number of diagnoses in each patient according to disease groups.** Each patient can have multiple diagnoses, so there are more diagnoses than patients. S1 Appendix shows the number of patients who have one or more diagnoses at the same time according to disease groups. Finally, it can see that the sum of the totals is equal to the sum of the diagnoses. (DOCX)

## Author Contributions

**Conceptualization:** Juan Carlos Andreu-Ballester.

**Formal analysis:** Juan Carlos Andreu-Ballester, Aurelio Pons-Castillo, Antonio González-Sánchez, María José Cano.

**Investigation:** Juan Carlos Andreu-Ballester, María José Cano.

**Methodology:** Juan Carlos Andreu-Ballester, Antonio González-Sánchez.

**Validation:** Juan Carlos Andreu-Ballester.

**Writing – original draft:** Juan Carlos Andreu-Ballester, Carmen Cuéllar.

**Writing – review & editing:** Juan Carlos Andreu-Ballester, Antonio Llombart-Cussac, Carmen Cuéllar.

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
