## [Decision Letter · Decision Letter 0]

22 Jul 2021

PONE-D-21-19890

Lymphopenia in hospitalized patients and its relationship with severity of illness and mortality.

PLOS ONE

Dear Dr. Andreu-Ballester,

Thank you for submitting your manuscript to PLOS ONE. After careful consideration, we feel that it has merit but does not fully meet PLOS ONE’s publication criteria as it currently stands. Therefore, we invite you to submit a revised version of the manuscript that addresses the points raised during the review process.

We look forward to receiving your revised manuscript.

Kind regards,

Aleksandar R. Zivkovic

Academic Editor

PLOS ONE

Reviewers' comments:

Reviewer #1: This is an interesting manuscript that explores an under-appreciated aspect of routine laboratory studies in patients. It is a welcome addition to the literature.

General comments: The manuscript at times does not quite make sense in English. I struggled to understand some parts of the paper. There are also several grammatical and spelling errors (e.g. in the abstract the first line says "mayor" when I believe the intent is "major").

Abstract: Rewrite the second sentence in the background. The current wording is confusing. There are also other spelling/grammatical errors (e.g. "mean time to dead" should probably be "mean time to death")

Introduction:

-In the first paragraph it mentions SIRS and states that includes criteria such as thrombocytopenia and neutrophilia. These are manifestations of sepsis but are not included in the SIRS definition. I would also include other criteria for sepsis (qSOFA, SOFA) in the introduction as well.

-The second paragraph needs rephrasing as well. SIRS in it's current definition is well defined but the accuracy and clinical utility are called into question. I believe the intent the authors are trying to convey with this sentence is that sepsis itself is difficult to diagnose.

-I recommend moving the definitions for SOI and ROM to the introduction from discussion.

Methods:

-Were these all patients or only adult patients?

Results:

-Again would revise some of the language to make the text more clear

-Table 2 is a bit confusing. From my understanding each admission gets one diagnosis as the number of cases equals the number of admissions captured. It would be helpful to explain how they were coded as such given that many patients should have multiple reasons for hospitalization (i.e. having both a respiratory and circulatory system issue).

-If feasible it would be interesting to compare lymphopenia in infectious diagnosis vs non-infectious diagnoses

-For Table 2 what does the acronym "DD" stand for?

-On page 8 why wwas 20 chosen as the days of lymphopenia?

-On page 8 I am unsure what the first sentence of the third paragraph is trying to convey. I believe more days of lymphopenia are linked with worse prognosis but I am unsure as the figure is unlabeled.

-In table 3 I think it would be beneficial if the significance was reported in leukopenia and neutropenia in the "bivariate analysis" given their inclusion in the multivariable analysis

Discussion:

-The first paragraph of page 10 needs to be revised heavily. I would also consider a citation of why lymphocytes are suppressed in sepsis

-The definitions of APR-DRG, SOI, and DRG should be moved to the introduction

-Why is lymphopenia during hospitalization more associated with mortality than lymphopenia on admission as show on page 8 and figure 2?

-I am not sure that the phrase "very significant direct relationship" can be used in this context on page 10. I would discuss this with a statistician to determine the appropriate language.

-On page 11 I do not think you can make the claim that "lymphopenia is clearly related to infectious pathology". Unless I missed it, the results do not specifically compare infectious vs non-infectious diagnoses so the authors do not know lymphopenia is more associated with sepsis.

Figures:

-Figure 1 needs the axes labeled on both x and y axes. It also needs a short explanation of what each table is describing

-Figure 2c needs to have the axes labeled and also an explanation of what it is conveying

-Figures 2D and 2E need to have the axes labeled and an explanation of what it is trying to convey

Overall the data presented is interesting. My main issues with the manuscript is the clarity and language, all I believe can be addressed with further revision.

Reviewer #2: Dear authors, this is a very interesting manuscript. I have some suggestions before publication:

1) In all tables you only provide information of the overall population and of the group of interest . However, to better understand the results it is important to provide the comparison group ( non limphopenia, patients who survived etc...) I presume that the p-value in these tables refer to the univariate logistic regressions. I believe that it would be more statistically sound to provide the univariate comparisons as descriptive statistics also for categorical variables (noun and %) using fisher test or qui- square test. Please add these comparisons to the statistical plan in the methods section and to the tables in the results section.

2)Please explain in the cox model which variable were included and provide (as supplement) the HR and CI for each adjusted variable.

3) Please explain further appendix S1. It is not clear to me what it represents for example 1 means that the patient only had one diagnosis within the disease subgroup?

4) The figures are very illustrative but please provide the individual p values of each comparison in figure 1. In figure 2 D and E I suggest that you provide the non limphopenia group as well and to compare both groups (limpohpenia in admission vs non limphopenia in admission and limphopenia during hospital stay and non limphopenia during hospital stay) providing a p value.

5) the manuscript needs English editing . for example in the abstract and in the introduction it is written mayor instead of major surgery . Also some sentences appear incomplete.

6. PLOS authors have the option to publish the peer review history of their article (what does this mean?). If published, this will include your full peer review and any attached files.

Reviewer #1: No

Reviewer #2: **Yes: **Elisa Gouvêa Bogossian

---

## [Author Response · Author response to Decision Letter 0]

1 Aug 2021

Response to Reviewers

Reviewers' comments:

Reviewer #1: This is an interesting manuscript that explores an under-appreciated aspect of routine laboratory studies in patients. It is a welcome addition to the literature.

General comments: The manuscript at times does not quite make sense in English. I struggled to understand some parts of the paper. There are also several grammatical and spelling errors (e.g. in the abstract the first line says "mayor" when I believe the intent is "major")

Reply: The manuscript has been thoroughly revised. We have tried to correct spelling and grammatical errors.

Abstract: Rewrite the second sentence in the background. The current wording is confusing. There are also other spelling/grammatical errors (e.g. "mean time to dead" should probably be "mean time to death")

Reply: 

The second sentence in the background has been corrected: “All these pathologies are clinically expressed by the so-called Systemic Inflammatory Response Syndrome which does not include lymphopenia as a defining criteria”.

“Dead” has been replaced by “death”

Introduction:

-In the first paragraph it mentions SIRS and states that includes criteria such as thrombocytopenia and neutrophilia. These are manifestations of sepsis but are not included in the SIRS definition. I would also include other criteria for sepsis (qSOFA, SOFA) in the introduction as well.

Reply: The sentence has been replaced by:

“which includes a number of clinical and laboratory criteria such as leukocytosis and leukopenia, but does not include lymphopenia”

-The second paragraph needs rephrasing as well. SIRS in it's current definition is well defined but the accuracy and clinical utility are called into question. I believe the intent the authors are trying to convey with this sentence is that sepsis itself is difficult to diagnose.

Reply: The sentence has been replaced by:

“There is a need for a consensual definition of SIRS and sepsis, since there is sometimes difficulty, with the current criteria, to define pathological cases with SIRS caused by infection, since it is not easy to identify the infectious focus in some situations”.

-I recommend moving the definitions for SOI and ROM to the introduction from discussion.

Reply: Definitions of SOI and ROM have been described in the introduction.

Methods:

-Were these all patients or only adult patients?

Reply: Pediatricians treat patients up to 14 years of age. The sentence has been replaced by: “Patients older than fourteen years were admitted from January 1, 2016, to December 31, 2019,….

Results:

-Again would revise some of the language to make the text more clear

Reply: The English edition has been corrected

-Table 2 is a bit confusing. From my understanding each admission gets one diagnosis as the number of cases equals the number of admissions captured. It would be helpful to explain how they were coded as such given that many patients should have multiple reasons for hospitalization (i.e. having both a respiratory and circulatory system issue).

Reply: The following text has been explained in Table 2:

“The diagnoses in each MDC correspond to a single organ system or cause, based on the main diagnosis on admission, therefore there is only one MDC per hospitalization episode” 

-If feasible it would be interesting to compare lymphopenia in infectious diagnosis vs non-infectious diagnoses

Reply: We have added in results the relationship of lymphopenia with non-infectious pathology, as well as with sepsis and septic shock. 

“In patients without infectious pathology, the incidence of lymphopenia is 34.6% (n = 12,876 / 37,167) vs 52.2% of lymphopenia in patients with infectious pathology (n = 11,016 / 21,093); ORlymphopenia without infectious pathology = 0.66 (0.63 - 0.65), p <0.001 " ORlymphopenia with infectious pathology = 2.06 (1.99 – 2.14), p <0.001.

There were a total of 6,208/58,260 (10.7%) patients with sepsis, 3,528 (56.8%) of which presented lymphopenia vs 2,680 (43.2%) who had lymphopenia in the group without sepsis, OR=2.1 (1.9 – 2.2), p<0.001.

The patients with Septic Shock were 515/58.260 (0.9%), of which 446 (86.6%) had lymphopenia vs 69 (13.4%) of the patients with Septic Shock who did not have lymphopenia, OR=9.5 (7.3 – 12.2), p<0.001.”

-For Table 2 what does the acronym "DD" stand for?

Reply: DD stands for Diseases and Disorders. In table 2 the acronym “DD” has been changed to “Diseases and Disorders”.

-On page 8 why was 20 chosen as the days of lymphopenia?

Reply: The first 20 days have been chosen because the number of patients with more than 20 days of admission is very low and therefore the results would not reach statistical significance. 

-On page 8 I am unsure what the first sentence of the third paragraph is trying to convey. I believe more days of lymphopenia are linked with worse prognosis but I am unsure as the figure is unlabeled.

Reply: Figure 2C shows how the percentage of deaths progressively increases according to the number of analyzes with lymphopenia that the patient has during their admission. Thus, as can be seen in Fig. 2C, 8.1% of patients who have had only one analysis with lymphopenia on admission or during admission die vs 30.3% of patients who have ten analyzes with lymphopenia during their admission. It has been explained in Figure 2C. 

-In table 3 I think it would be beneficial if the significance was reported in leukopenia and neutropenia in the "bivariate analysis" given their inclusion in the multivariable analysis

Reply: Table 3 has been modified to analyze the variables by Two-tailed Fischer's exact test, logistic regression and Cox regression. The method has been added in Statistical analysis.

Discussion:

-The first paragraph of page 10 needs to be revised heavily. I would also consider a citation of why lymphocytes are suppressed in sepsis

Reply: We have justified the possible causes of immunoparalysis associated with sepsis. Three bibliographic citations have been added

“The most probable cause of this transient phenomenon of immunoparalysis with decreased of lymphocytes has been attributed to increased apoptosis of this cells [19, 20]”

-The definitions of APR-DRG, SOI, and DRG should be moved to the introduction

Reply: Definitions of SOI and ROM have been described in the introduction

-Why is lymphopenia during hospitalization more associated with mortality than lymphopenia on admission as show on page 8 and figure 2?

Reply: We believe that lymphopenia during hospitalization is more associated with mortality than lymphopenia on admission because the percentage of patients who die increases progressively according to the number of analyzes performed with lymphopenia during their hospital admission, as shown in Figure 2C. These results have been commented in the discussion section.

-I am not sure that the phrase "very significant direct relationship" can be used in this context on page 10. I would discuss this with a statistician to determine the appropriate language.

Reply: After consulting a statistician as suggested by the reviewer we have removed the word “very”. The correct expression is “significant direct relationship”.

-On page 11 I do not think you can make the claim that "lymphopenia is clearly related to infectious pathology". Unless I missed it, the results do not specifically compare infectious vs non-infectious diagnoses so the authors do not know lymphopenia is more associated with sepsis.

Reply: As already indicated above, we have added in results the relationship of lymphopenia with non-infectious pathology, as well as with sepsis and septic shock. 

“In patients without infectious pathology, the incidence of lymphopenia is 34.6% (n = 12,876 / 37,167) vs 52.2% of lymphopenia in patients with infectious pathology (n = 11,016 / 21,093); ORlymphopenia without infectious pathology = 0.66 (0.63 - 0.65), p <0.001 " ORlymphopenia with infectious pathology = 2.06 (1.99 – 2.14), p <0.001.

There were a total of 6,208/58,260 (10.7%) patients with sepsis, 3,528 (56.8%) of which presented lymphopenia vs 2,680 (43.2%) who had lymphopenia in the group without sepsis, OR=2.1 (1.9 – 2.2), p<0.001.

The patients with Septic Shock were 515/58.260 (0.9%), of which 446 (86.6%) had lymphopenia vs 69 (13.4%) of the patients with Septic Shock who did not have lymphopenia, OR=9.5 (7.3 – 12.2), p<0.001.”

Figures:

-Figure 1 needs the axes labeled on both x and y axes. It also needs a short explanation of what each table is describing

Reply: Figure 1 has been better explained. 

-Figure 2c needs to have the axes labeled and also an explanation of what it is conveying

Reply: The axes in figure 2C have been labeled and conveniently explained. 

-Figures 2D and 2E need to have the axes labeled and an explanation of what it is trying to convey

Reply: Figures 2D and 2E have been labeled and better explained. 

Overall the data presented is interesting. My main issues with the manuscript is the clarity and language, all I believe can be addressed with further revision.

Reviewer #2: Dear authors, this is a very interesting manuscript. I have some suggestions before publication:

1) In all tables, you only provide information of the overall population and of the group of interest. However, to better understand the results it is important to provide the comparison group (no lymphopenia, patients who survived etc...) I presume that the p-value in these tables refer to the univariate logistic regressions. I believe that it would be more statistically sound to provide the univariate comparisons as descriptive statistics also for categorical variables (noun and %) using fisher test or qui- square test. Please add these comparisons to the statistical plan in the methods section and to the tables in the results section.

Reply: Table 3 has been modified to analyze the variables by Two-tailed Fischer's exact test, logistic regression and Cox regression. The method has been added in Statistical analysis.

2) Please explain in the cox model which variable were included and provide (as supplement) the HR and CI for each adjusted variable.

Reply: In the logistic regression analysis and COX regression, all the analytical variables have been introduced into the model (Table 3), and it´s explained at the bottom of the table.

3) Please explain further appendix S1. It is not clear to me what it represents for example 1 means that the patient only had one diagnosis within the disease subgroup?

Reply: Each patient can have several diagnoses so there are more diagnoses than patients. The appendix S1 shows the number of patients who have 1 or more diagnoses at the same time according to groups of diseases. Finally, it can be seen that the sum of the totals coincide with the diagnoses.

4) The figures are very illustrative but please provide the individual p values of each comparison in figure 1. In figure 2 D and E I suggest that you provide the non limphopenia group as well and to compare both groups (limphopenia in admission vs non limphopenia in admission and limphopenia during hospital stay and non limphopenia during hospital stay) providing a p value.

Reply: Figures 1 and 2 show the percentage of patients with lymphopenia in each subgroup (minor, moderate, major, and extreme). If we have not misunderstood, the percentage of patients without lymphopenia will be the rest of the percentage until reaching 100%, so we believe that it would be a figure with too many lines that would be confusing. Statistical analysis has been explained.

“Figure 2. Kaplan-Meier plots illustrating Survival probability following lymphopenia on admission (A) and lymphopenia during hospital stay (B) in peripheral blood of admitted patients. Time is expressed in days (mean). Log-Rank test (p-value). Percentage of dead patients according to the number of analyzes performed with lymphopenia on admission or during admission (C). Percentage of patients with lympophenia on admission and during hospital stay according to four subclases of Severity of illness (SOI) (D) and Risk of Mortality (ROM) (E). Chi square Pearson test, p<0.001. Pearson's R: Interval-by-interval, p<0.001. Spearman correlation: ordinal by ordinal, p<0.001 (D and E panels)”

5) The manuscript needs English editing, for example in the abstract and in the introduction, it is written mayor instead of major surgery. Also some sentences appear incomplete.

Reply: We have made a new edition in English.

---

## [Editor Report · Decision Letter 1]

3 Aug 2021

Lymphopenia in hospitalized patients and its relationship with severity of illness and mortality.

PONE-D-21-19890R1

Dear Dr. Andreu-Ballester,

We’re pleased to inform you that your manuscript has been judged scientifically suitable for publication and will be formally accepted for publication once it meets all outstanding technical requirements.

Kind regards,

Aleksandar R. Zivkovic

Academic Editor

PLOS ONE

---

## [Editor Report · Acceptance letter]

5 Aug 2021

PONE-D-21-19890R1 

Lymphopenia in hospitalized patients and its relationship with severity of illness and mortality. 

Dear Dr. Andreu-Ballester:

I'm pleased to inform you that your manuscript has been deemed suitable for publication in PLOS ONE. Congratulations! Your manuscript is now with our production department. 

Kind regards, 

on behalf of

Dr. Aleksandar R. Zivkovic 

Academic Editor

PLOS ONE